# Research Progress on Construction of Lutein-Loaded Nano Delivery System and Their Improvements on the Bioactivity

**Yongqiang Ma [1], Tingting You [1,*], Jing Wang [2], Yan Jiang [3] and Jichao Niu [1]**

1   School of Food Engineering, Harbin University of Commerce, Harbin 150028, China
2   School of Light Industry, Harbin University of Commerce, Harbin 150028, China
3   School of Public Health, DALI University, Dali 671000, China
*   Correspondence: ytt_happiness@163.com

**Abstract:** Lutein belongs to the diverse group of pigments known as oxygenated carotenoids, also known as phytochrome and macular pigment, demonstrating excellent biological activity. However, its application is limited due to the difficulty of dissolution, poor stability, and low bioavailability. To solve these problems, delivery systems are considered to be one of the most promising choices. These delivery systems can improve the physical, chemical, and biological properties of lutein to a certain extent. Moreover, the system can also be adapted to the needs of production in our daily life. In this paper, the construction of lutein-loaded nano delivery systems and their influences on the bioactivity of lutein were reviewed based on previous researchers. The main materials were classified, and assistant substances, basic parameters, and properties were collected. The mechanisms were analyzed in terms of enhancing cellular uptake, improving bioavailability, and achieving targeted delivery. These results show that different materials have their own characteristics. This review aims to provide references for the production and application of lutein in the food industry.

**Keywords:** lutein; delivery system; biological activity; targeting



## 1. Introduction

Lutein is a kind of fat-soluble pigment, belongs to oxygenated carotenoids, also known as phytochrome and macular pigment [1]. The structural formula and the 3D configuration of lutein can be seen in Figure 1 [2]. It is widely recognized by consumers due to its unique golden color used as a coloring agent. Meanwhile, the excellent bioactivity and safety of lutein have attracted many companies to use it as one of the preferred raw materials for "clean label" series products [3].

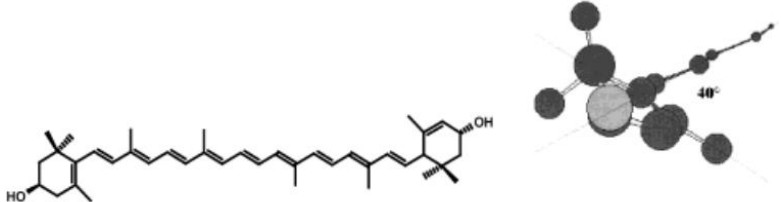

**Figure 1.** The structural formula of lutein [2]. Copyright 2003, Journal of Agricultural and Food Chemistry. All rights reserved.

With the changes in modern lifestyle, life rhythm, and population composition, there has been an increase in the prevalence of central neurodegenerative diseases, geriatric diseases, mental disease [4–6], and so on. Despite the rapid development of related diagnostic and therapeutic technologies in recent years, the pathogenesis and cure of these diseases have not been fully revealed. In 2008, Tan et al. [7] suggested that a daily intake of about 6 mg of lutein could reduce the risk of cataracts and macular degeneration, which

provides ideas from the perspective of nutritional intervention of lutein for chronic diseases. In addition, a large number of studies have shown that lutein has strong antioxidant and anti-inflammatory properties [8], which give lutein rich bioactivity. Several experiments in vivo and in vitro, as illustrated in Table 1, demonstrated that lutein has the function of neuroprotection, retina protection, hypolipidemic, and so on. However, the difficult dissolution, poor stability, and low bioavailability have limited the application of lutein. At present, a large number of experiments have shown that a delivery system may be an excellent measure.

**Table 1.** Researches on the biological activity of lutein.

| Biological Activity | Cell or Model | Method | Delivery System |
| --- | --- | --- | --- |
| retina protection [9,10] | blue light damage model in male rats | intragastric | free |
| | HCE-F | incubate | free |
| inhibited tumor cell proliferation [11–13] | HEPG2 | incubate | free |
| | HeLa cell/MDCK | incubate | free (purity > 92.2%) |
| | MDA-MB-157/MCF-7 | incubate | free (purity > 99%) |
| hypolipidemic [14] | HEPG2 | incubate | free (purity > 97%) |
| anti-listeria monocytogenes infection [15] | RAW 264.7 macrophage | incubate | free (purity > 98%) |
| | EGD model in female rats | subcutaneous | |
| neuroprotection [16] | PD model by rotenone-induced in drosophila | feed | polymer-based |
| relieved FM [17] | FM model in female rats | intravenous | lipid-based |

EGD, Elevated Glucocorticoid Disease; PD, Parkinson's Disease; FM, Fibromalgia.

Delivery systems are often used in the field of pharmaceutical research [18,19] to improve the stability of drugs [20], reduce toxic side effects [21], and achieve targeted drug delivery [22]. In recent years, the concept of a "novel nano delivery system" has been proposed in the food industry and widely used in the development and production of special medical foods [23], health foods [24], and functional foods [25]. Delivery systems usually consist of substances and bioactive compounds. Polymers [26], natural products [27–30], bionic biomaterials [31–33], metallic materials [34,35], magnetic materials [36,37], silicon-based materials [38–40], etc. are commonly used as substances. To choose proper substances, one should consider their structures, physicochemical properties, and suitability for bioactive components. Special attention should be paid to the substances that present cytotoxicity when used in the food industry [41]. Currently, main bioactive components delivered are polyphenols [42], natural pigments [43,44], alkaloids [45], radiopharmaceuticals [46], mineral drugs [47], etc. Common properties of these substances are poor water solubility, easy degradation, or high toxic side effects. On the one hand, delivery systems protect bioactive components from production and application environment. On the other hand, their aim is to reduce metabolic burden or organic damage caused from the toxic side effects of bioactive compounds. For example, Figure 2a shows the delivery system for curcumin-loaded with exosomes as main substances, which is a kind of bionic biomaterial that more easily penetrates the blood–brain barrier to act on the brain [48]. Figure 2b shows the delivery system for lycopene-loaded with polycaprolactone and dimethyldioctadecylammonium bromide as main substances which can enhance cellular uptake and may be an effective tool for anti-cancer therapy [49]; Figure 2c shows the delivery system for

palm oil-loaded with chitosan, picolin emulsion, and sodium alginate as main substances, which provides a method for the immobilization of bioactive compounds [50]. In this paper, the construction of lutein-loaded nano delivery systems and the mechanism of their effects on the bioactivity of lutein were reviewed based on the previous researchers. We explored the main substances, assistant substances, basic parameters and properties of the lutein-loaded nano delivery system. The influences were analyzed in terms of enhancing cellular uptake, improving bioavailability, and achieving targeted delivery. This review aims to provide references for the production and application of lutein in the food industry.

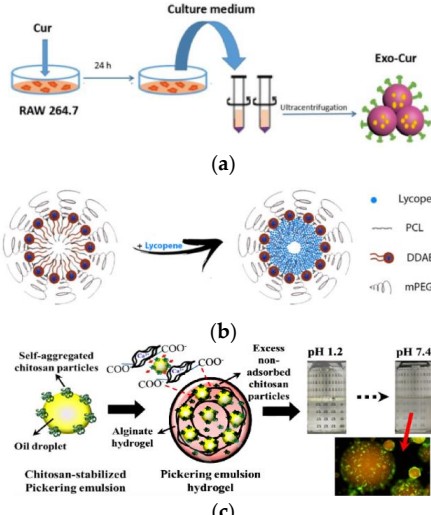

**Figure 2.** Process of delivery system for bioactive compounds loaded. (**a**) Delivery system of curcumin-loaded [48] Copyright 2019, Nanoscale. All rights reserved; (**b**). Delivery system of lycopene-loaded [49] Copyright 2021, Journal of Drug Delivery Science and Technology. All rights reserved; (**c**) Delivery system of palm olein-loaded [50] Copyright 2020, Food Hydrocolloids. All rights reserved.

## 2. Construction of Lutein-Loaded Nano Delivery Systems

Currently, lutein-loaded nano delivery systems are classified into nano emulsion, nano particle, nano capsule, nano fiber, nano crystal, etc. They can be observed by scanning electron microscopy (SEM) or transmission electron microscopy (TEM) in the shape of spheres, flakes, vesicles, fibers, or capsules, as shown in Figure 3. Lutein was loaded by bonding interaction or physical distribution. In this section, basic parameters and properties of the systems were classified according to categories of main materials (base). Moreover, the "substances" mentioned in the text refer to main materials other than lutein or related bioactive compounds.

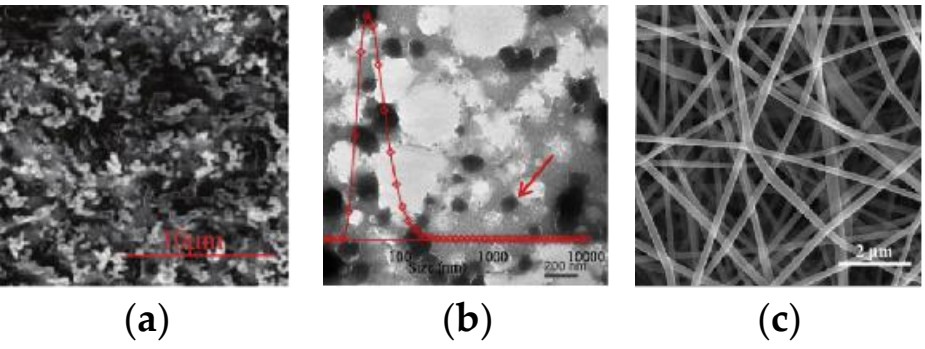

**Figure 3.** *Cont.*

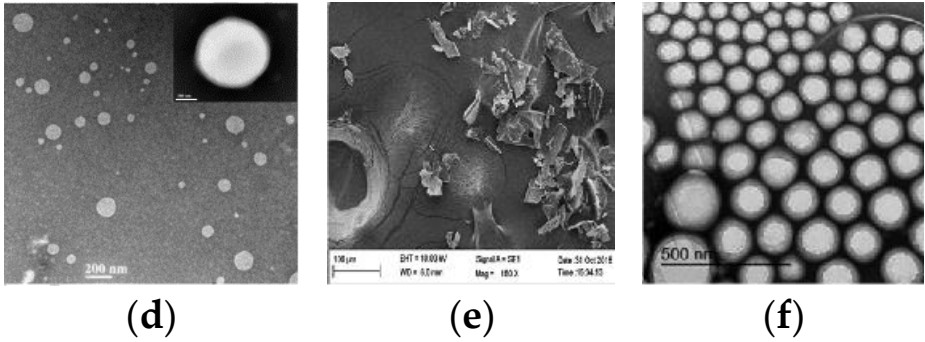

**Figure 3.** Morphology of lutein-loaded nano delivery system observed by SEM or TEM. (**a**) Nano emulsion photographed by cryo-SEM [51] Copyright 2018, Food & Function. (**b**) Nano chitosomes photographed by TEM, like spherality-shape [52] Copyright 2016, Food Hydrocolloids. (**c**) Nano fiber photographed by SEM [53] Copyright 2019, Pharmaceutics. (**d**) Bulk vesicular photographed by TEM, like spherality-shape [54] Copyright 2017, Food Research International. (**e**) Irregular flaky nanoparticles photographed by SEM, like elongated rod-shape [55] Copyright 2021, Food Chemistry. (**f**) Nano particles photographed by TEM, like spherality-shape [56] Copyright 2019 Nanomedicine.

### 2.1. Polymers

Polymers were used as main materials to construct lutein-loaded nano delivery system as shown in Table 2, mainly including PLGA, PEG, PVP, etc.

The systems have shown that the diameter of average particle (D) was from dozens to hundreds nanometer, the Z-potential (Z) were all negative and the absolute values were more than 20 mV mostly, $\lambda$ and the polydispersity index (PDI) were all about 0.2. As we all know, these parameters play a decisive role on stability of the systems. For example, a higher negative electrical transition from the carboxyl terminus makes the absolute value of Z larger which presented better optical and thermal stability [57]. Meanwhile, the assistant technology could be one of the methods to make the values of D smaller and the systems more uniform so as to achieve better properties, such as semi-continuous high-throughput electrofluidic-mediated mixing technique (EM-NP), small batch interfacial instability ultrasonication (II-S) [58].

Encapsulation efficiencies (EE) of lutein were about 70% in the systems. The factors affecting the EEs include preparation method, distribution, or solubility mainly. The precipitation method was effective in promoting the EEs, e.g., the system prepared by PCL has the EE of 99.51% [59]. Moreover, lycopene-loaded systems prepared by the methods have higher EEs [60]. Nevertheless, the EEs of the systems prepared by PS-PEO and PCL-PEG were extremely low [48] with the value about 4%, which was predicted to correlate with compatibility and molecular interactions through Hansen solubility parameters (HSPs) $\delta$, HSPs distance Ra, and Flory–Higgins parameters Xsp, etc.

In general, polymers as main substances showed good stability when stored at low temperatures. Moreover, most systems can realize the release, which could be important in enhancing bioavailability. However, polymer-based substances may be toxic [53]. Hence, the safe dose should be paid more attention to.

### 2.2. Natural Products

Natural products are the ingredients extracted from plants, animals, microorganisms or their metabolites, etc. They are very popular because of their varieties, wide sources, and high safety. Natural products are greatly different in structure, physicochemical property, and composition, such that the systems exhibit their own characteristics (Table 3).

**Table 2.** Study on polymers as main materials for lutein-loaded nano delivery system construction.

| Materials | | Basic Parameters | | | | | Properties |
|---|---|---|---|---|---|---|---|
| Base | Others | D (nm) | Z (mV) | PDI | EE (%) | Morphology | |
| PLGA PVA [61] | TM | 210.6 ± 3.3 | −6.7 ± 0.3 | 0.119 ± 0.007 | 87.6 ± 1.4 | nanoparticle sphere-shape | improved stability (i) degradation rate of 26% in 4 °C (5 weeks) realized release |
| PLGA PEG [57] | B | 208.0 ± 3.38 | −27.2 ± 2.04 | 0.206 ± 0.016 | 74.56 ± 10.25 | nanoparticle sphere-shape | realized sustaining and controlled release no obvious cytotoxicity in $C_{50 \mu g/mL}$ |
| PLGA PEG [62] | F | 188.0 ± 4.06 | −26.0 ± 1.27 | 0.202 ± 0.009 | 72.87 ± 7.22 | nanoparticle sphere-shape | improved stability (i) No obvious change on D, Z and PDI in 4 °C/25 °C (60 days) realized burst release (initial), then sustaining and controlled release (later) |
| PLGA [63] | lecithin | 140 ± 6 | −43 | 0.186 | 90 ± 2 | nanoparticle sphere-shape | improved solubility and stability (i) increased solubility by 86% than free L in 50 °C (48 h) (ii) retention rate of 26% in UV (24 h) realized burst release (initial), then sustaining and controlled release (later) |
| PLGA [64] | D-Ta | 222.9 ± 1.2 | −32.4 ± 3.9 | 0.131 ± 0.023 | 96.2 ± 2.7 | nanoparticle sphere-shape | realized sustaining release exhibited pseudoplastic behavior |
| PS-PEO [58] | – | 26.3 ± 0.290 | – | – | 4.46 ± 0.94 | nanomicelle | – |
| PS-PEO [58] | – | 32.1 ± 0.231 | – | – | 2.09 ± 0.49 | nanomicelle | – |
| PCL-PEG [58] | – | 24.7 ± 0.300 | – | – | 14.08 ± 3.25 | nanomicelle | – |
| PCL-PEG [58] | – | 25.8 ± 0.244 | – | – | 9.86 ± 1.24 | nanomicelle | – |
| PCL [59] | MCT Tw80 | 191.9 ± 3.24 | −5.14 ± 2.22 | 0.11 ± 0.02 | 99.51 | nanocapsule sphere-shape | improved stability (i) no obvious change on Z, pH, color and retention rate in 4 °C/25 °C (90 days) |
| PVA [53] | SA | 240~340 | – | – | 91.9 ± 2.58 | nanofiber bar-shape | improved hydrophilic (i) increased contact angle to 78.2 ± 0.37° realized sustained slow-release exhibited cytotoxicity |
| PLL [65] | CHOL Tw80 PC | 367.1 ± 7.94 | −27.9 ± 0.68 | 0.400 ± 0.036 | 92.93 ± 5.39 | nanoliposome Sphere-shape ellipsoidal-shape | realized sustaining release (i) release rate of 51.26 ± 3.33% in SGF (20 h) (ii) release rate of 70.32 ± 1.42% in SIF (20 h) improved stability (i) degradation rate of 30.95 ± 3.33% in SGF (12 h) and that of 27.67 ± 1.42% in SIF (12 h) |
| PVP [66] | Tw80 | ≈200 | – | – | – | nanocapsule sphere-shape | improved solubility (more than 43-fold) |

PLGA, poly (lactic-co-glycolic acid); PVA, poly vinyl alcohol; TM, Trehalose monohydrate; B, biotin; F, folate; PEG, polyethyleneglycol; D-Ta, D-Trehalose; PS-PEO, poly (styrene-b-ethylene oxide); PCL, polycaprolactone; MCT, medium chain triglycerides oil; SA, sodium alginate; PLL, polylysine; CHOL, cholesterol; Tw80, tween-80; PC, phosphatidylcholine; PVP, polyvinylpyrrolidone; C, concentration; SGF, simulated gastric fluid; SIF, simulated intestinal fluid.

**Table 3.** Study on natural products as main materials for lutein-loaded nano delivery system construction.

| CAT | Substances | | Basic Parameters | | | | | Properties |
|---|---|---|---|---|---|---|---|---|
| | Base | Others | D (nm) | Z (mV) | PDI | EE (%) | Morphology | |
| PROTEIN | ZP [67] | GlcN | <200 | – | – | 89.60 | nanoparticle sphere-shape | improved solubility<br>exhibited aggregation |
| | DHZP [68] | Try | <125 | Abs > 30 | <0.4 | – | | improved stability<br>(i) no obvious change on D and Z |
| | DHZP [68] | Fla | <300 | Abs > 30 | <0.4 | – | nanoparticle sphere-shape | unstability<br>(i) significant change on D and Z<br>(ii) be easy to show aggregation |
| | DHZP [68] | Tw80 | <125 | Abs > 30 | < 0.4 | – | | improved stability<br>(i) no obvious change on D, Z and color |
| | DHZP [69] | Pep | 112.24 ± 1.56 | −25.6 ± 1.06 | 0.039 ± 0.008 | 93.82 ± 2.82 | nanoparticle sphere-shape | improved stability<br>(i) degradation rate of 29.10 ± 0.806% in SGF (12 h)<br>(ii) degradation rate of 25.98 ± 0.932% in SIF (12 h) |
| | ZP [70] | SSPS | ≈200 | ≈0.15 | 0.039 | >80 | nanoparticle sphere-shape | improved stability<br>(i) retention rate of 96.27 ± 2.80% in 15 days<br>(ii) better pH value and saline solution stability<br>good redispersibility<br>no cytotoxicity |
| | ZP [71] | TS | 213.1 | −25.6 | <0.2 | 92.91 | nanoparticle sphere-shape | improved stability<br>(i) no obvious change on D and PDI in pH = 4~9<br>(ii) no obvious change on D and PDI in $C_{NaCl0-100mM}$<br>(iii) no obvious change on D and PDI in 37 °C/55 °C/80 °C (2 h)<br>(iv) retention rate of more than 90% in 12 days<br>significant change on D and PDI in pH = 2~3 |
| | ZP [56] | Trehalose | 220.1 ± 6.2 | pH 6 + 15.7 ± 1.5 | 0.197 ± 0.024 | 95.9 ± 3.2 | nanoparticle sphere-shape | exhibited pseudoplastic behavior |
| | ZPDP [72] | – | 297.7 ± 7.55 | −22.5 ± 1.48 | 0.458 ± 0.026 | 90.32 ± 3.56 | nanoparticle sphere-shape | good redispersibility<br>improved solubility (more than 12-fold)<br>improved stability<br>(i) degradation rate of 32.32 ± 1.36% in SGF (7 h)<br>(ii) degradation rate of 21.22 ± 0.84% in SGF (6 h)<br>realized release<br>(i) release rate of 21.22 ± 0.84% in SGF (7 h)<br>(ii) release rate of 34.08 ± 1.48% in SGF (7 h) |
| | ZP [73] | PC F127 | 216.5 ± 29 | −47.6 ± 1.6 | <0.3 | 83 ± 5.8 | nanoparticle sphere-shape | improved stability<br>(i) no obvious change in 4 °C (30 days)/UV (10 h)<br>(ii) release rate of 19.38% and 42.67% in PBS (0 h) and PBS (24 h), respectively |

**Table 3.** *Cont.*

| CAT | Substances | | Basic Parameters | | | | | Properties |
|---|---|---|---|---|---|---|---|---|
| | Base | Others | D (nm) | Z (mV) | PDI | EE (%) | Morphology | |
| | CSCA [74] | CS | $331 \pm 22$ | $\approx +30$ | $<0.2$ | $43.82 \pm 5.69$ | nanoparticle sphere-shape | improved stability (i) no obvious change on morphology and PDI in 25 °C (5 weeks) (ii) retention rate of $49.3 \pm 6.1\%$ |
| | SBSA [75] | GA CMC | $242.20 \pm 0.50$ | $-30.40 \pm 0.70$ | $0.27 \pm 0.01$ | $83.95 \pm 0.98$ | nanoparticle sphere-shape | improved thermal and storage stability |
| | LDHRP [76] | | $170 \pm 2$ | $-34 \pm 0$ | $0.39 \pm 0.03$ | $>90$ | nanoparticle sphere-shape | improved stability (i) higher degree of Hydrolysis showed better stability |
| | MDHRP [76] | Try | $160 \pm 10$ | $-35 \pm 2$ | $0.36 \pm 0.02$ | $>90$ | nanoparticle sphere-shape | |
| | HDHRP [76] | | $143 \pm 3$ | $-38 \pm 2$ | $0.32 \pm 0.03$ | $>90$ | nanoparticle sphere-shape | |
| | (G)α(s1)-ICP | | $206.44 \pm 2.66$ | $-35.14 \pm 1.77$ | – | – | nanoemulsion | enhanced affinity with L by (G) α(s1)-ICP improved stability |
| | (G)α(s1)-IICP | A-Gal [77] | $206.56 \pm 2.66$ | $-34.80 \pm 2.46$ | – | – | nanoemulsion | |
| | (B) CP | | $205.94 \pm 2.9$ | $-37.50 \pm 1.86$ | – | – | nanoemulsion | |
| | SF [78] | CS | $\approx 8.9$ | pH4 $\approx +28$ | – | 16.0 | nanoparticle sphere-shape | improved stability (i) release rate of $5.0 \pm 0.4\%$ in dialysis of 4 °C (32 h) (ii) retention rate of 74.1% in 20 °C (7 days) |
| | BSA [79] | CA DEX $V_E$ | $\approx 220$ | – | – | – | nanoemulsion sphere-shape | improved stability (i) inhibited aggregation (ii) no obvious change on D and PDI in 4 °C/37 °C (pH = 7, 15 days) |
| | BSA [80] | FUC | $304.4 \pm 10.7$ | $-60.7 \pm 5.0$ | 0.372 | – | nanoemulsion | improved stability (i) no obvious change on D in 4 °C/25 °C/55 °C (ii) degradation rate of 82%, 79% and 36% in 4, 25, and 55 °C respectively worse thermal stability |
| | PPI [81] | DEX | pH = 7 $269 \pm 36$ pH = 4.6 $396 \pm 15$ | pH = 7−$11.2 \pm 0.45$ pH = 4.6−$0.83 \pm 0.18$ | pH = $70.76 \pm 0.42$ pH = 4.6 $1.79 \pm 0.13$ | – | nanoemulsion | improved stability (i) no obvious change on D and PDI in 4~37 °C (pH = 7, 30 days) (ii) no obvious change on D and PDI in divalent ion ($C_{0\sim100Mm}$) and monovalent ion significant change on D and PDI in pH = 4.6 |

**Table 3.** *Cont.*

| CAT | Substances | | Basic Parameters | | | | | Properties |
|---|---|---|---|---|---|---|---|---|
| | Base | Others | D (nm) | Z (mV) | PDI | EE (%) | Morphology | |
| | CP [81] | DEX | pH = 7<br>125 ± 1<br>pH = 4.6<br>123 ± 1 | pH = 7<br>−12.67 ± 0.82<br>pH = 4.6<br>−0.2 ± 0.22 | pH = 7<br>0.13 ± 0.00<br>pH = 4.6<br>0.13 ± 0.03 | – | nanoemulsion | improved stability<br>(i) no obvious change on D and PDI in 4~55 °C<br>(pH = 7, 30 days) and<br>55 °C (pH = 4.6, 30 days)<br>(ii) no obvious change on D and PDI in divalent ion<br>($C_{0~100Mm}$) and monovalent ion<br>significant change on D and PDI in 4~37 °C (pH = 4.6) |
| | CP [82] | DEX<br>Res<br>MCT<br>GSO | <150 | – | – | – | nanoemulsion | improved stability under pH value and saline solution<br>improved color stability |
| | CP [83] | DEX | 118.5 ± 7.56 | – | 0.340 ± 0.02 | 97.16 ± 1.25 | nanomicelle | improved stability in divalent ion solution and SGF |
| | LF [51,84] | MCT | 251.1 | +22.6 | – | – | nanoemulsion | – |
| | WPI [85] | – | 202 ± 9.7 | – | 0.29 ± 0.02 | – | nanoemulsion | improved stability<br>(i) no obvious change on D and morphology in<br>4 °C (4 weeks) |
| | PMP [85] | – | 209 ± 3.3 | – | 0.27 ± 0.02 | – | nanoemulsion | exhibited stratification |
| STARCH | β-CD [86] | CO<br>F-68<br>Span20 | 91.7 ± 0.8 | −33.1 ± 0.1 | – | 95.1 ± 1.4 | nanoemulsion | improved stability<br>lower cytotoxicity |
| | OMS [87] | – | 198~235 | pH 6 − 4 | – | – | nanoliposome<br>sphere-shape | improved stability<br>(i) no obvious change on D in<br>5 days or $C_{NaCl5-100mM}$<br>(ii) no obvious change on D and Z in pH = 3~7 |
| | OSA-SGC [88] | – | 187.25 | – | – | 89.79 | nanomicelle<br>particles | exhibited aggregation with the change of pH value and<br>saline solution concentration<br>realized controlled release |

**Table 3.** *Cont.*

| CAT | Substances | | Basic Parameters | | | | | Properties |
|---|---|---|---|---|---|---|---|---|
| | Base | Others | D (nm) | Z (mV) | PDI | EE (%) | Morphology | |
| CHITOSAN | CS [89–91] | OA Tw80 SA | 98 ± 5 | +38 ± 4 | 0.27 ± 0.01 | 90 | nanoparticle sphere-shape | no cytotoxicity |
| | CS [90] | OA SA | 10−150 | +45 ± 5 | 0.174 ± 0.201 | – | nanoparticle sphere-shape | improved solubility and thermal stability realized burst release (initial), then sustaining and controlled release (later) exhibited higher adhesion of 80 ± 2% |
| | CS [92] | PLGA | <150 | – | – | >80 | nanoparticle sphere-shape | no cytotoxicity realized sustaining and stable release no burst release improved light and thermal stability |
| | CS [93] | TPP PC | 65.2 ± 3.2 | +47.01 ± 0.86 | 0.03 ± 0.00 | 90 ± 1 | nanoparticle sphere-shape | improved stability (i) release rate of 3.2%, 18% in SGF (2 h) and SGF (10 h), respectively (ii) release rate of 68% in SIF (2 h) no cytotoxicity |
| | CS [94] | DS | ≈400 | +46 | – | 60~76 | nanoparticle sphere-shape | improved stability good adsorption capacity |
| | LMWC [95] | – | 80~600 | – | – | 85 ± 1 | nanocapsule sphere-shape | – |
| OTHER NATURAL PRODUCTS | STE [55] | – | 165 ± 2 | −38.33~−30.90 | 0.09 | 94.07~72.19 | nanoparticle sheet-shape | improved stability |
| | OA LOA [96] | Tw20 | 110 ± 8 | +36 ± 2 | 0.271 | – | nanoliposome sphere-shape | improved solubility (726-fold) improved stability (i) no obvious change on D and Z in 4 °C (30 days) (ii) retention rate of 94 ± 4% |
| | SC [97] | – | 234.01 ± 3.4 | −36.56 ± 1.5 | 0.123 ± 0.028 | – | nanoemulsion | improved stability (i) no obvious change on D and Z in 4 °C (30 days), $C_{1:5}$ (30 days) or 60 °C (30 days) (ii) retention rate of 91.51 ± 2.83%, 86.68 ± 1.91% in $C_{1:5}$ (30 days) or 60 °C (30 days) exhibited unstability with the change of pH value and saline solution concentration |
| | CFG [98] | – | 162 | −8.23 | – | 86 | nanoemulsion | improved stability (i) no obvious change on D, Z, isomerization and degradation rate |

**Table 3.** *Cont.*

| CAT | Substances | | Basic Parameters | | | | | Properties |
|-----|------|--------|----------|--------|-----|--------|------------|------------|
| | **Base** | **Others** | **D (nm)** | **Z (mV)** | **PDI** | **EE (%)** | **Morphology** | **Properties** |
| | CO [99] | V$_E$($\alpha$) WPI | 68.8 ± 0.3 | – | <0.2 | 80.7 ± 0.8 | nanoemulsion sphere-shape | improved stability (i) no obvious change on D in 5 °C/20 °C/40 °C (28 days) significant change on L's content and color values in 5 °C/20 °C/40 °C (28 days) |
| | SC [100] | – | 231.8 ± 1.6 | >30 | 0.155 ± 0.015 | – | nanoemulsion | exhibited color's unstability with the change of temperature |
| | SC [101] | DEX | 138.25 ± 0.5 | Abs < 10 | – | – | nanoemulsion | inhibit flocculation exhibited adverse charge with pH value of solution |
| | FO [102] | – | 167.5 ± 0.793 | −34.2 ± 0.50 | 0.172 ± 0.016 | 88.5 ± 4.21 | nanoliposome sphere-shape | accelerated release velocity |
| | SBP [103] | FG WPI DTAB Lac | 100~400 | −25~30 | – | – | nanoemulsion | inhibited aggregation improved stability |

CAT, catalogue; ZP, zein; GlcN, Glucosamine; DHZP, dehydrogenation zein; Try, trypsin; Fla, flavorase; Pep, pepsin; SSPS, soluble soybean polysaccharide; TS, tea saponin; DP, derivatized peptides; F127, Pluronic F127; CSCA, *Camellia* seed cake albumin; CS, chitosan; SBSA, *Stauntonia brachyanthera* seed albumin; GA, gum Arabic; CMC, carboxymethylcellulose; LDHRP, low dehydrogenation rice protein; MDHRP, medium dehydrogenation rice protein; HDHRP, high dehydrogenation rice protein; A-Gal, arabinogalactan; G$\alpha$ (s1)-ICP, goat $\alpha$ (s1)-I casein; G$\alpha$ (s1)-IICP, goat $\alpha$ (s1)-II casein; B-CP, bovine casein; SF, soybean ferritin; BSA, bovine albumin; CA, chlorogenic acid; DEX, dextran; V$_E$, vitamin E; FUC, fucoidan; PPI, pea protein; Res, resveratrol; GSO, grape seed oil; LF, lactoferrin; WPI, whey protein isolate; PMP, polymeric whey protein isolate; CD, cyclodextrins; CO, corn oil; OSA, octenyl succinic anhydride; F-68, pluronicf-68; OMS, octenyl modified starch; SGC, short glucan chains; OA, oleic acid; TPP, tripolyphosphate; DS, dextran sulfate;LMWC, low molecular weight chitosan; STE, Stevia; LOA, linoleic acid; Tw20, tween-20; SC, sodium caseinate; CFG, corn fiber gum; FO, fish oil; SBP, sugar beet pectin; FG, fish gelatin; DTAB, dodecyltrimethylammonium bromide; Lac, laccase.

### 2.2.1. Protein-Based Substances

Zein is often used as a kind of common substance to construct delivery systems for various bioactive components, such as lycopene, β-carotene, and curcumin [99,104,105]. It has been often used in the lutein-loaded nano delivery systems. Besides, other proteins have been also applied in the systems, such as *Camellia* seed cake albumin [74], *Stauntonia brachyanthera* seed albumin [75], rice protein [76], goat/bovine casein [77], soybean ferritin [78], bovine serum albumin [79], pea protein [80], lactoferrin [51,84], whey protein isolate, polymeric whey protein isolate [85], etc.

In the systems, high concentration of salt solution may trigger particle aggregation to form large particle size and destabilize the system. Due to the isoelectric point of protein, the stability of the systems was also influenced greatly by pH values. For example, ZP shows instability at pH 5~6, CP tends to aggregate particles at pH 4.6, and LF tends to produce precipitation at pH 6.0. Currently, there are three methods which may be valid to avoid the isoelectric point:

- The first is the enzymic method, such as GlcN, Try, Pep, Fla, or ZPDP, etc. A study [76] reported the molecular binding mechanism of RP prepared by enzyme decomposition. It was found that DHRP with higher enzymatic hydrolysis degree have stronger activity and stability. This may be related to protein denaturation or structural changes.
- In the second method, other substances could be paired with protein to prepare composite substances, such as ZP with SSPS [70] or TS [71], CP with A-Gal [77], and CSCA with CS [74]. These composites substances utilize charge interactions to stabilize the protein over a wider pH range.
- The third method involves the addition of DEX to prepare a Maillard reactive substance (MRPS) [79–83], which is more effective in inhibiting aggregation and producing a spatial site barrier effect. For example, Yong et al. prepared the BSA-CA-DEX for lutein-loaded, which formed spherical nanoemulsion with uniform distribution and an average particle size of about 220 nm with good stability. Thereby, the stability of the systems was improved by the above factors.

### 2.2.2. Starch-Based Substances

Starch is renewable, environmentally friendly, and offers low-pollution as a kind of biomass material. Exploring the high utilization of starch is one of the hot research topics in the context of carbon peaking and carbon neutrality. Currently, starch-based substances for the systems are amphiphilic polymers with OSA modified SGC [88], CD [86], OMS [87], etc. The systems have good stability, but the stability mechanism is different from that of polymers. Mostly, the charge number, acid-base environment, or particle size produced little effect on the stability. They were determined due to the formation of inclusion complexes with CD, the competition under multi-component conditions, and the spatial site resistance induced by the starch moiety.

### 2.2.3. Chitosan-Based Substances

Chitosan belongs to the groups known as alkaline polysaccharides and constructs the systems with positive charge. The absolute values are all greater than 30 mV. Properties of positive potential and good adhesion make chitosan-based substances stand out under the conditions of SGF and SIF. Various forms of release can be achieved, such as controlled release, sudden (initial) release, sustained release, continuous stable release, and so on. The addition of lipids, such as oleic acid (OA), Tw80 [89–91], and PC [93] can reduce the D values to further improve the stability of the systems. It is particularly worthwhile to mention that the chitosan-based substance used for the systems is almost non-cytotoxic. In conclusion, chitosan is a kind of ideal substance for food delivery.

### 2.2.4. Other Natural Products-Based Substances

Stevia [55], a natural sweetener, was extracted from Asteraceae, and often used as a sugar substitute. STE has a six-membered ring π system, a central ring structure, etc.,

which interacts with lutein to produce hydrogen bonds. The combined effect of C-H-π interaction forces, van der Waals forces, and intermolecular forces ensures the stability of the system. Oleic acid and linoleic acid [96] are unsaturated fatty acids presented in animal and plant oils fatty acids. The solubility of OA-LOA for lutein-loaded was increased by 726 times compared to that of free lutein, which provides solutions of difficult solubility. Caseinate exits usually with the form of sodium salt. The SC for the system [97] has good thermal stability, but it is more sensitive to the pH values. Corn fiber gum [98], corn oil, fish oil [102], and beet pectin [103] were also used as materials for the systems. To some extent, all of these could improve the stability of lutein. In addition, some natural products also play an important role in the systems. For example, $V_E$ and GSO can enhance chemical stability and resveratrol can reduce fading.

In summary, natural products-based substances were used for lutein-loaded nano delivery systems have been widely applied. However, the systems are more complex, and the formation mechanism is often difficult to discern. It is recommended that raw materials should be of high purity. When you need a match, the materials may be no more than two as far as possible.

### 2.3. Lipids

Lipid-based substances for a lutein-loaded nano delivery system construction are shown in Table 4 with a wide range of particle size distribution. The smallest particle size among the lipid-based substances is 12.7 ± 0.7 nm, which is prepared by isopropyl myristate, triethyl triacetate and Tween 80 with ultrasound assistance [106]. Cosby et al. [58] prepared the systems by II-S and EM-NP methods with particle sizes of 22.6 ± 0.178 nm and 23.0 ± 0.204 nm, respectively. Nano emulsion prepared by egg yolk phospholipids [52] as main substances had particle sizes not exceeding 100 nm with a core-shell structure. The formation of such small D values was analyzed to be possibly related to the lipid, most of which have particle sizes between 100 nm and 500 nm. However, few systems have large particle sizes. For example, the emulsion prepared by tea polyphenol palmitate and xanthan gum have D values of tens of microns [107].

At the same time, they have better stability to be applied in the case of a wide temperature range. One study about the systems prepared by LAE-Tw80 [101] showed stable systems under the temperature of 4 °C, 25 °C, and 37 °C. Moreover, the system prepared by EGCG-β-lg [108] shows the better degradation rate of lutein of only 12.8% when stored at 4 °C for 30 days. Advance technology can also be useful to the stability of lutein-loaded systems. The nano liposome was prepared by PC with the supercritical $CO_2$ extraction technology, which improved the stability of the system under the temperature of 308 K, 313 K, 318 K and the pressure of 10~15 MPa [109]. Based on the experimental phenomena above, some researchers have discussed the mechanism, which may be related to the values of emulsification activity index (EAI), emulsion stability index (ESI) [110], the concentration of lipids, or the properties of lipids [111,112].

It is worth noting that most lutein-loaded nano delivery systems prepared by lipid-based substances tend to have good release effects. This phenomenon was analyzed to be most likely related to their lipophilic structures [113].

**Table 4.** Study on lipids as main materials for lutein-loaded nano delivery system construction.

| Substances | | Essential Parameters | | | | | Properties |
|---|---|---|---|---|---|---|---|
| Base | Others | D (nm) | Z (mV) | PDI | EE (%) | Morphology | |
| EGCG [108] | β-lg | 138.0 (25:1) | − | <0.2 (25:1) | − | nanoemulsion sphere-shape | improved stability (i) no obvious change on D in 4 °C (30 days) (ii) degradation rate of 12.8% in 4 °C (30 days) |
| GM [114] | LHP P-188 | 118.5 ± 1.02 | −25.84 ± 2.45 | 0.136 ± 0.017 | 94.43 ± 1.08 | Solid lipid nanoparticle sphere-shape | improved stability (i) enhanced thermal-resistant, light-resistant and oxygen-resistant stability of 4.42-fold, 3.41-fold and 3.21-fold higher than free L, respectively (ii) enhanced the PAPP (1.52-fold than free L) realized sustained slow-release |
| LAE [110] | − | ≈357 | +86.53 ± 5.12 | − | − | nanoemulsion | improved stability (i) no obvious change on D in 4 °C/25 °C/37 °C, better in 4 °C/25 °C EAI of 14.608 ± 0.367 and ESI of 0.954 ± 0.022 exhibited phase separation under SGF |
| Tw80 [110] | − | ≈428 | −25.53 ± 1.72 | − | − | nanoemulsion | improved stability (i) no obvious change on D at 4 °C/25 °C/37 °C, better at 4 °C/25 °C EAI of 14.664 ± 0.336 and ESI of 0.980 ± 0.016 |
| SDS [110] | − | ≈289 | −95.20 ± 1.97 | − | − | nanoemulsion | significant change on D at 4 °C/25 °C/37 °C, better at 4 °C EAI of 15.096 ± 0.352 and ESI of 0.983 ± 0.017 |
| PC [109] | − | 145 ± 54 | − | − | 97.8 ± 1.2 | nanoliposome vesicle | realized release |

**Table 4.** *Cont.*

| Substances | | Essential Parameters | | | | | Properties |
|---|---|---|---|---|---|---|---|
| **Base** | **Others** | **D (nm)** | **Z (mV)** | **PDI** | **EE (%)** | **Morphology** | |
| PC [109] | – | 65 ± 33 | – | – | 91.9 ± 2.9 | nanoliposome | improved stability (i) no obvious change on D and EE in 308 K/313 K/318 K and $F_{10\sim15MPa}$ |
| PC [94] | – | 109.8 | +16.9 | <0.2 | 100 | nanogel irregular shape | improved stability under STF cytotoxicity |
| PC [54] | – | 147.6~195.4 | −54.5~−61.7 | 56.7~97.0 | - | nanoparticle | - |
| MCT [115] | Tw80 | ≈200 | ≈0.23 | – | – | nanoemulsion | improved stability (i) no obvious change on D and PDI in 20 °C (4 weeks) significant change on color values in 20 °C (4 weeks) |
| IPM [106] | TT Tw80 | 12.7 ± 0.7 (NE-5) | – | 0.07 ± 0.03 (NE-5) | – | nanoemulsion sphere-shape | improved stability (i) no obvious change on D realized release rate of 66.3 ± 13.2% |
| EYPC Tw80 [52] | CS | <100 | <20 | <0.25 | 85~90 | nanoemulsion sphere-shape | improved thermal-resistant stability realized controlled release |
| TW80 [116] | – | 123.1 ± 0.3 | – | 0.155 ± 0.008 | 93.16 | nanoparticle | – |
| | – | 136.9 ± 8.4 | – | 0.136 ± 0.001 | 91.36 | nanoparticle | – |
| TPGS MCT [117] | – | 254.2 | 265 | 0.29 | – | nanoemulsion | – |
| Precirol ATO5 [118] | 18-04KF68 | 134 ± 8 | −36.3 ± 2.9 | 0.18 ± 2.02 | – | nanoliposome disk-shape | improved stability under SGF realized release |

EGCG, epigallocatechin gallate; β-lg, β-lactoglobulin; GM, glycidyl methacylate; LHP, linoleic acid hydroperoxides; P-188, Poloxamer-188; LAE, polyoxyethylene fatty acid ester; SDS, sodium dodecyl sulfate; IPM, isopropyl myristate; TT, triethyl triacetate; EYPC, egg yolk phospholipids; TPGS, D-alpha-Tocopheryl polyethylene glycol succinate; F68, pluronicf-68; STF, simulated tear fluid.

In summary, polymers, natural products, lipids, etc. can be used as materials for the construction of lutein-loaded nano delivery systems and have their own characteristics, which are summarized as follows:

1.  D values of nano delivery systems are greatly affected by the preparation techniques, such as EM-NP, II-S, SFE-CO$_2$, HPH, UT-assisted, etc.
2.  The materials, such as CS, LAE, PC, MCT, and some proteins (ZP, CP, etc. when in a specific pH environment), can be used to prepare nano delivery systems with positive charges.
3.  The stability of nano delivery systems includes physical stability, chemical stability, storage stability, digestive stability, etc. The determination mainly examines the D values, potentials, color values, isomerization and degradation rates, retention rates and release rates of lutein, etc.
4.  The concentration and ratio of main materials, the pH value, the temperature, and the duration of the systems located in the environments will have an impact on their stabilities.
5.  Compounds can improve the stability of the systems, but they can also lead to unexplained phenomena and uncertainty. Therefore, three substances or more are not to be recommended to use simultaneously.

## 3. Improvements on the Bioactivity of Lutein Using Nano Delivery System

### 3.1. Enhanced the Cell Uptake

The amount of cellular uptake of bioactive components determines the magnitude of the function. Several studies have shown that the cellular uptake of bioactive components occurred mainly through the form of endocytosis, internalization degree, or transfer. For example, the systems prepared by stevia realized the cellular uptake of lutein via lipid raft-mediated endocytosis. In another study, the systems were prepared by the PLGA-PEG-F [62] and showed enhanced cellular uptake in neuroblastoma cells. It was found that the addition of folic acid as receptor-mediated endocytosis occurred during cellular uptake. Similar results were also verified in the experiment of the PLGA-PEG-B [57]. Except for the receptor action, polymer-based materials for the systems were well absorbed when passing through lipid bilayer of the cells into cytoplasm and nucleus with better internalization efficiency. Meanwhile chitosan-based materials presented with higher internalization degree. This is due to its hydrophilic surface and the electrostatic action that occurs between the surface and cell membrane.

### 3.2. Improved the Bioavailability

Bioavailability describes the ratio of remaining content of drug after digestion to the initial content, which is using maximum concentration (C$_{max}$), area under the drug-time curve (AUC) as evaluation indexes. The causes of low bioavailability include materials interference (i.e., unstable physicochemical properties, poor water solubility and imbalanced ratio of complex), processing conditions, unknown biotransformation effects, and so on.

One of the factors affecting the bioavailability of lutein is the degree of free fatty acid release. Hao et al. [119] prepared the NaCAS-ALG nano particle with lutein using electrostatic complexation. The large proportion of digested lipid phase led to the formation of more mixed micelles. The amount of free fatty acid release and bile salt binding in the mixed micelle phase directly determined the ability to dissolve lutein. Thus, the bioavailability was increased to 51.27%, which was higher than that of NaCAS nano emulsion with lutein and free lutein.

Secondly, different natural products as materials for the systems also have an effect on the bioavailability of lutein. STE can effectively prevent lutein accumulation and make it evenly dispersed in the brush border of intestinal epithelial cells to promote absorption [55]. Alginate can reduce particle aggregation and increase the contact surface area of oil droplets to dissolve lutein in micelles [90]. Then, the lutein was absorbed by the small intestine. Polysaccharide can protect lutein from pepsin and achieve controlled release under the

condition of SGF, while it is dissociated into SIF; zein with TS can increase bioavailability of lutein by 67.17% compared to free lutein [71].

Thirdly, the addition of phospholipids may improve the bioavailability of lutein due to hydrolysis of phospholipids under the action of intestinal phospholipase. A study compared the bioavailability of PLGA-PL/PLGA with lutein and free lutein. The results of SGF digestion assay after 4 h in vitro showed that the system prepared by PLGA-PL with lutein had the intestinal mucosa absorptivity greater than 80%, significantly higher than PLGA with lutein and free lutein.

Except for the degree of free fatty acid release, the special properties of natural products and the action of phospholipids have better solubility. Moreover, it was found that lutein in the systems exhibited amorphous form. Thus, lutein is more easily dispersed in the digestive solution, which improves the bioavailability.

### 3.3. Realized the Targeting Delivery

#### 3.3.1. Eye Targeting

Several studies have shown that lutein is preferentially absorbed by the liver and fat, whereas we expect higher ocular levels. Chitosan-based material has shown a dose-dependent increase in the eyes of mice [91]. In another cell permeability test on rabbit cornea, it was observed that lutein nano liposome gradually entered the cornea and reached a steady state within 90 min. The release rate was maintained balance from 90 min to 240 min to realize eye targeting. A more detailed study was about distribution of lutein in 11 tissues throughout the eyes [57]. Thus, it is to be concluded that the pathway of lutein into the eyes is probably through conjunctiva, sclera, choroid, and then retina. The results clearly demonstrate that nano delivery system can deliver lutein to the eyes. However, the duration of action is very short, and the effect is not good. Thus, the materials used need to be optimized in terms of adhesion, dispersibility, and release to apply for long duration.

#### 3.3.2. Brain Targeting

Research on brain targeting constitutes one of hot topics in the scientific field. Solutions for delivering bioactive compounds into brain are still not well constructed due to the blood–brain barrier mainly. Lutein has neuroprotective effects, i.e., can prevent degenerative diseases (AD, PD, FM, etc.) that may be caused by neurological damage. Therefore, lutein could be a medium used for constructing the systems into brain, then to be applied for other nutrients. At present, several materials have been found to be useful in brain targeting:

- The first is polymer-based material, using a nano delivery system to enhance cognitive performance in mice [66].
- The second is a kind of cationic liposomes capable to restore monoamine content in cortical tissue and cortical electroencephalogram signals [17].
- Besides, transnasal administration is also a means of brain targeting which could deliver bioactive compounds more efficiently. By this method, an approximately 4.4-fold increase compared to free lutein was observed in a test [92].

Although the above systems have led to changes of indicators in the brain, the pity is that the mechanism of lutein crossing the BBB, the pathway into the brain, and the duration of action have not been fully revealed.

#### 3.3.3. Others

Lutein in liver and plasma is also studied by researchers. The system prepared by CS-SA with lutein was prepared to increase lutein content in plasma and liver of normal mice and diabetic mice by 3.1-fold, 7.3-fold and 2.7-fold, 3.4-fold [89]. Compared to lutein micelles, a kind of polymer material added to phospholipids in the systems leads to an increase of 3.91-fold and 2.89-fold in plasma and liver [63]. Low molecular weight chitosan mainly as material for the system also can enhance lutein content in plasma and liver [95].

In summary, polymers, chitosan, and lipids as main materials can be applied to construct the systems for realizing targeted delivery of lutein. The applications of materials

for targeted delivery to brain, eye, liver and plasma to exert biological activity have been shown in Figure 4. Smaller particle size, biocompatible materials, and stable systems contribute to lutein crossing the blood–retina barrier and blood–brain barrier to some extent. Moreover, contact time and duration in the eye and the brain can be prolonged, while absorption and utilization are also promoted.

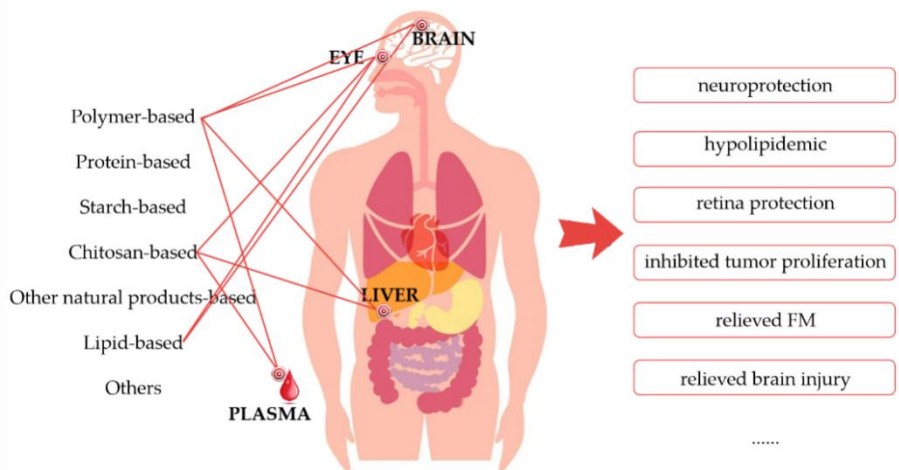

**Figure 4.** Application of materials for targeted delivery to brain, eye, liver and plasma to exert biological activity.

## 4. Conclusions

Novel nano delivery systems are promising and significant for applications in the fields of food and medicine. In this paper, research progress of lutein-loaded nano delivery systems was reviewed in two major aspects. Firstly, main materials of lutein-loaded nano delivery systems construction were classified into four parts, including polymers, natural products, lipids, and others. The systems were concluded from basic substances (base), assistant substances (others), essential parameters (D, Z, PDI, EE and morphology), and properties (stability, adhesion, toxicity or release). The second is that improvements in the bioactivity of lutein by using the systems above are analyzed. This section explained important roles of the systems in terms of enhancing cellular uptake, improving bioavailability, and achieving targeted delivery. Considering the need for further optimization of the process and in-depth investigation on the mechanism of action, future research may be carried out as follows:

(1) Regarding theory, there were already rich research bases. However, theoretical research still needs to be further developed, such as optimization patterns of formation environment, which are necessary to provide cost-effective preparation solutions in order to be more competitive, and analysis of the mechanism, especially on BRB and BBB.

(2) Regarding production, most of the relevant research is still in the laboratory stage and few pilot studies and industrialization are available. This will be a great challenge because the system is influenced by the environment.

(3) Finally, regarding application, currently, marketed products with lutein are mainly in eye protection and the active ingredients are mostly lutein only. Meanwhile, there are even fewer commercial applications for improving immunity and promoting neuroprotection. This will be a commercial opportunity in the future.

In conclusion, we should refer to the research methods and experiences from the field of medicine so as to construct lutein-loaded nano delivery systems with excellent characteristics. More systematic research protocols, accurate detection methods, and advanced technology will be necessary to explore the mystery of lutein in human metabolism.

**Author Contributions:** Conceptualization, Y.M., Y.J. and T.Y.; validation, T.Y. and J.W.; resources, T.Y.; data curation, T.Y.; writing—original draft preparation, T.Y. and Y.J.; writing—review and editing, T.Y., J.W. and J.N.; supervision, Y.M. and J.W.; funding acquisition, Y.M., J.W. and T.Y. All authors have read and agreed to the published version of the manuscript.

**Funding:** This research work was supported by the Heilongjiang Province Applied technology research and development project (GA20B301), the project of the National Natural Science Foundation of China (52002099) and the Harbin Science and Technology Bureau project (CY2020JH020105).

**Institutional Review Board Statement:** Not applicable.

**Informed Consent Statement:** Not applicable.

**Data Availability Statement:** The authors confirm that the data supporting the findings of this study are available within the article.

**Acknowledgments:** The authors would like to thank the support of the project of the National Natural Science Foundation of China, the Heilongjiang Province Applied technology research and development project and the Harbin Science and Technology Bureau project for the supports on the paper.

**Conflicts of Interest:** The authors declare no conflict of interest.

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
