# Peer review of "Research Progress on Construction of Lutein-Loaded Nano Delivery System and Their Improvements on the Bioactivity"

_coatings, doi:10.3390/coatings12101449_

Round 1

Reviewer 1 Report

The review article has been adequately structured, ranging from the technological aspect of the Lutein-loaded Nano Delivery System to the reports of biological activity on different tissues and cell lines.

References are properly used.

Just recommend to improve the tables 1 (column of properties) and 2 (Title). In both cases, please clarify the codes of materials used. Many initials appear that are difficult to identify. It is better to put the full names to make reading easier.

Reviewer 2 Report

Dear Editor

Thank you for this opportunity to review the paper by Yongqiang Ma et al. In this paper, the authors reviewed the preparation methods of lutein-loaded nano delivery systems and investigation their biological activities. The manuscript is not well prepared, with many typos (freshman mistakes) that demonstrate poor accuracy. The subject is not critically described, frequently a list of examples summarized in Tables is reported rather than a comparison between methods with advantages, disadvantages, peculiarities and so on. A review article is a critical description of methods and not just a list of entries. Below please find a list of extra comments:

1. The title of manuscript should be improved.

2. Make sure that any abbreviations that are used are defined somewhere in the manuscript.

3. Please insert chemical structure of lutein to the manuscript.

4. The quality of all figures in the manuscripts is very low. Please improve them.

5. The title of section 3 must be improved.

6. There is no point of the release mechanism, biological activity of prepared lutein-loaded nano delivery systems, just reporting a general table without a comparative discussion….

Overall I consider the manuscript unsuitable for publication and I recommend rejection.

Reviewer 3 Report

186 should be written without italics - family Asteraceae

190 watch the space - [84]are unsaturated fatty acid

319 should be written in italics - 3.3.2. Brain targeting; 3.3.3. Others

The article uses many abbreviations that are not explained. I suggest a separate section in the article explaining all the abbreviations. Also, you can explain them in the main part of the article and below each table.

The references are not written according to the rules of the journal

Author Response

请看附件

Round 2

Reviewer 2 Report

Dear Editor
The paper was revised according to the reviewer’ comments.
In its current state it is ready for publication in your journal.
Best regards